# ViewCraft3D: High-Fidelity and View-Consistent 3D Vector Graphics Synthesis

**Chuang Wang** [1*] **Haitao Zhou** [1*] **Ling Luo** [2†] **Qian Yu** [1†]

[1]Beihang University    [2]Ningbo University

[*]Equal contribution    [†]Corresponding author

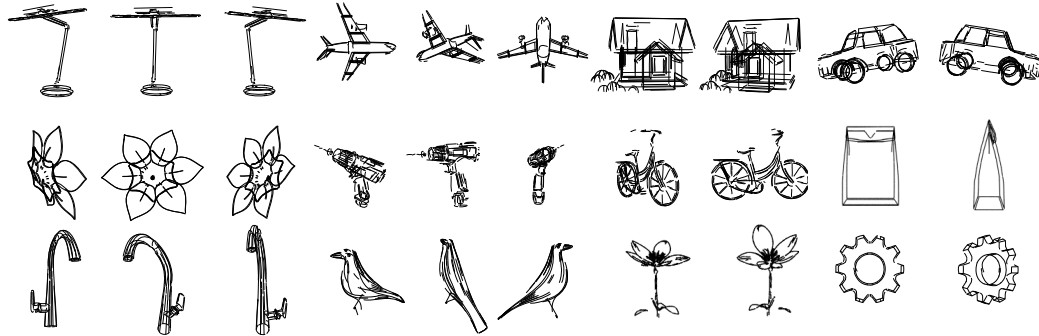

Figure 1: We propose *ViewCraft3D (VC3D)*, a method to generate 3D vector graphics from a single image. VC3D can leverage 3D prior knowledge to generate high-quality and view-consistent 3D vector graphics.

## Abstract

3D vector graphics play a crucial role in various applications including 3D shape retrieval, conceptual design, and virtual reality interactions due to their ability to capture essential structural information with minimal representation. While recent approaches have shown promise in generating 3D vector graphics, they often suffer from lengthy processing times and struggle to maintain view consistency. To address these limitations, we propose **ViewCraft3D** (VC3D), an efficient method that leverages 3D priors to generate 3D vector graphics. Specifically, our approach begins with 3D object analysis, employs a geometric extraction algorithm to fit 3D vector graphics to the underlying structure, and applies view-consistent refinement process to enhance visual quality. Our comprehensive experiments demonstrate that VC3D outperforms previous methods in both qualitative and quantitative evaluations, while significantly reducing computational overhead. The resulting 3D sketches maintain view consistency and effectively capture the essential characteristics of the original objects. Project page: `https://zhtjtcz.github.io/VC3D_page/`.

## 1 Introduction

Three-dimensional vector graphics offer a unique balance between abstraction and comprehensibility, using minimal line elements to convey complex spatial information. These economical representations have become integral to diverse computing applications, from improving immersive experiences in virtual environments to facilitating 3D shape retrieval and reconstruction tasks [54, 20, 19, 13, 53]. In virtual reality creation environments, 3D vector graphics serve as intuitive building blocks that allow artists to materialize spatial concepts directly within immersive spaces [56, 1, 55], bridging the gap between imagination and digital realization. Recent interactive sketching tools [2, 1, 57]

39th Conference on Neural Information Processing Systems (NeurIPS 2025).

have enhanced these creative capabilities by enabling direct manipulation in 3D space. Despite these advances, creating effective 3D vector graphics remains prohibitively difficult for non-specialists due to the intricate combination of spatial reasoning, technical interface skills, and artistic judgment required. This expertise barrier significantly limits widespread adoption and accessibility, highlighting the need for automated approaches that can generate high-quality 3D vector graphics without requiring users to have specialized training or artistic expertise.

Recent years have witnessed remarkable progress in 2D vector graphics generation. Works like CLIPasso [36] and CLIPDraw [7] pioneered the use of CLIP's visual-semantic understanding to guide vector graphics optimization. Building on these foundations, methods such as VectorFusion [11], DiffSketcher [45], and SVGDreamer [47] further leveraged diffusion models to achieve higher fidelity and controllability in vector graphics generation. Concurrently, the field of 3D content creation [49, 59, 15, 40, 37, 51] has been revolutionized by neural rendering techniques and generative models, making high-quality 3D asset creation in-

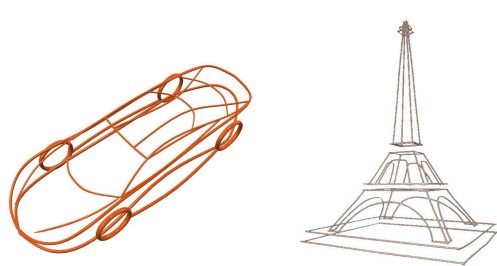

Figure 2: Examples of VR sketches [33].

creasingly accessible. The convergence of these advancements has catalyzed research in 3D vector graphics, with pioneering works like 3Doodle [5] and Diff3DS [58] demonstrating the feasibility of generating expressive 3D line drawings. These approaches have achieved impressive results in creating 3D vector graphics. However, existing methods predominantly rely on 2D generative priors—leveraging models like CLIP [28] and diffusion model [30] as supervision signals—while employing Score Distillation Sampling (SDS) [24] for optimization in 2D projection space rather than directly in 3D. These indirect approaches inherit a fundamental limitation of 2D SDS optimization: cross-view inconsistency, which constrains the ability of methods [24, 58]—where the same 3D element appears inconsistently from different viewpoints. Even with the use of more powerful pretrained models, these approaches often struggle to generate coherent 3D vector graphics that remain consistent across arbitrary viewpoints. For example, Diff3DS [58] employs MVDream [32] to tackle this issue, but the improvement is only partial. On the other hand, 2D priors from pretrained image generation models offer only conceptual-level guidance, lacking precise recovery of critical lines typically found in human-drawn 3D sketches, as illustrated in Figure 2. As a result, the generated outputs often suffer from messy strokes, missing details, and low structural fidelity.

To overcome these challenges, we propose **View**C**raft3D** (VC3D), a novel approach that leverages 3D priors for generating high-fidelity and view-consistent 3D vector graphics. Instead of relying on optimization using 2D priors [58, 5], our method is grounded in 3D geometric attributes within the 3D domain. This allows it to naturally inherit the cross-view consistency of the 3D object while faithfully preserving its spatial structure and geometric details, as illustrated in Figure 1. Specifically, we start by reconstructing a 3D mesh using a pre-trained image-to-3D model. Based on the resulting mesh, we identify salient regions in 3D space that capture the object's key structural features. We then perform point-level clustering using spatial proximity and orientation alignment. These clusters are subsequently fitted with 3D Bézier curves, and Chamfer Distance loss is used to ensure accurate geometric approximation. To further refine these vector graphics, we introduce a 3D score distillation sampling loss based on pretrained 3D generative models, which optimizes the Bézier curve parameters to enhance both visual quality and structural fidelity. This approach maintains view consistency by construction, as the optimization occurs directly in 3D space guided by 3D priors.

In summary, our contributions are threefold:

- We propose ViewCraft3D (VC3D), a novel framework for generating high-fidelity 3D vector graphics that leverages 3D priors rather than 2D projections;

- We develop a two-phase optimization approach combining geometric fitting with 3D-prior guided refinement, significantly improving visual quality.

- We conduct extensive experiments demonstrating that our approach outperforms existing methods in both view consistency and generation speed. The results suggest promising directions for future studies.

# 2 Related Work

## 2.1 2D Vector Graphics Generation

Early approaches in 2D SVG generation, such as CLIPasso [36] and CLIPDraw [7], utilized the visual-semantic understanding of the CLIP model [28] to guide vector optimization. Subsequent research introduced more sophisticated approaches, notably those employing diffusion models [8, 30, 6]. Work like VectorFusion [11], DiffSketcher [45], SVGDreamer [47], and SVGDreamer++ [46] demonstrated significant improvements in generation quality by employing Score Distillation Sampling [24]. This technique effectively transfers the generative capabilities of pixel-based models to the vector domain. LLM4SVG [43] attempts to use LLM to generate SVG, while Reason-SVG [41] introduces the idea of reinforcement learning to generate better SVG. In addition, SVGFusion [44] explored the use of the DiT architecture [23] to generate SVG. Furthermore, specialized approaches have been developed for specific applications. These include Word-as-image [10] for typographic design, CLIPascene [35] for scene sketching with varying abstraction levels, and VectorPainter [9] for stylized graphics synthesis.

More recently, efforts have focused on mitigating the computational cost associated with iterative optimization. Works based on autoregressive models, such as Iconshop [39], have demonstrated the potential for rapid generation, significantly reducing processing times. Concurrently, the adaptation of large language models (LLMs) for SVG generation has emerged as another promising research avenue, with works like LLM4SVG [42] and Chat2SVG [38]. And OmniSVG [50] attempts to employ Vision-Language Models (VLMs) as end-to-end multimodal SVG generators. Together, these recent advancements aim to ensure high-quality generation while paving the way for future extensions into 3D representations.

## 2.2 Recent Advances in 3D Content Generation

Recent years have witnessed remarkable progress in 3D content generation driven by diffusion-based approaches. Early works like Zero-123 [17] pioneered single-image view synthesis using geometric priors from diffusion models, while One-2-3-45 [16] extended this to generate full 360-degree textured meshes. Multi-view consistency became a focus with MVDream [32], which serves as an implicit 3D prior through multi-view image generation, and Wonder3D [18], which employs cross-domain attention for consistent normal and color generation. Recent innovations have further elevated capabilities: Unique3D [37] improved fidelity through multi-level upscaling, HunYuan3D [49, 59] achieved photorealistic quality, TripoSG [15] utilized triplane optimization with large-scale data, and Hi3DGen [51] enhanced geometric fidelity through normal bridging. These cutting-edge approaches primarily focus on generating complete 3D assets with textures and materials, while our work emphasizes the creation of 3D vector graphics that maintain characteristic abstractions and representational efficiency. By leveraging the 3D understanding embedded in these advanced models, particularly TripoSG's structural representations, we guide our vector optimization toward semantically meaningful and view-consistent results.

## 2.3 3D Vector Graphics Generation

Building upon both the 2D vector graphics techniques and recent 3D generation advances discussed above, 3D vector graphics generation has emerged as a promising research direction. These representations extend the fundamental advantages of 2D vector graphics while leveraging 3D generative capabilities to model complex spatial structures and depth information. This integration enhances their utility in diverse fields, including web development and digital art. In artistic contexts, works like DreamWire [27] and Fabricable 3D Wire Art [34] have showcased the potential of 3D vector graphics to create compelling, view-dependent visual effects, where the perceived objects change based on viewing angle. To harness these benefits and enable such advanced applications, the development of robust 3D vector graphics generation techniques has become a key research focus. Initial explorations in this area include 3Doodle [5], which pioneered a method for generating 3D vector graphics from multi-view images of the target object. Subsequently, Diff3DS [58] utilized the Score Distillation Sampling to produce 3D vector graphics conditioned on text or image input. Dream3DVG [52] leverages the optimization process of a 3D Gaussian Splatting [12] to establish a coarse-to-fine generation approach. However, a notable aspect of these current generative approaches is their predominant reliance on 2D view-specific loss for optimization. View-specific loss is computed independently per camera view, and gradients are aggregated across views during optimization. Without strong

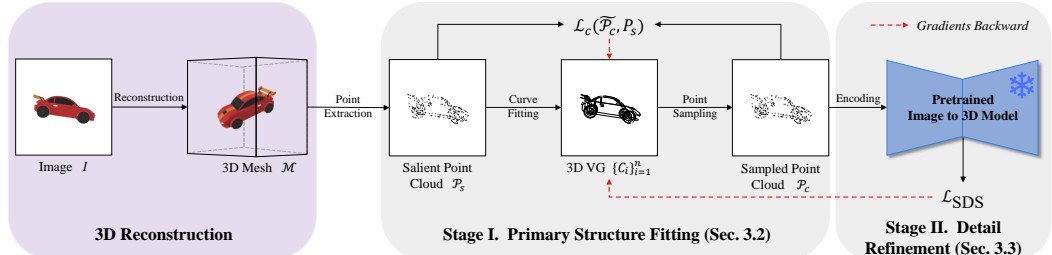

Figure 3: The overall architecture of the proposed method, showcasing the initial generation of 3D Vector Graphic (3D VG) from an input image and subsequent detail refinement using a pretrained image-to-3D model.

3D regularization (e.g., geometry priors or multi-view constraints), this results in locally optimal solutions per view that can conflict globally. While yielding impressive outcomes, this strategy may not fully exploit 3D spatial cues, potentially leading to challenges such as view inconsistency in the final 3D vector representations.

## 3 Methodology

### 3.1 Overview

In this section, we introduce **ViewCraft3D** (**VC3D**), an optimization based method that creates a 3D vector graphic $\mathcal{S}^{3D}$ based on an input image $I$. We define a 3D vector graphic $\mathcal{S}^{3D}$ as a set of 3D Bézier curves $\{C_i\}_{i=1}^{n}$. The curves are defined by a set of control points $\{P_{i,j}\}_{j=1}^{m}$, where $P_{i,j} \in \mathbb{R}^3$ is the $j$-th control point of the $i$-th curve.

Our method workflow is illustrated in Figure 3. It begins by reconstructing a 3D mesh $\mathcal{M}$ from a user-provided image $I$ using an image-to-3D model [15]. We then apply a two-stage process on the resulting mesh, consisting of Bézier curve fitting followed by detail refinement. The primary structure fitting stage identifies high-curvature regions in the reconstructed mesh, converts them into a point cloud, and fits Bézier curves to approximate these structures. The detail refinement stage re-initializes additional curves in regions overlooked during the first stage and optimizes them using Score Distillation Sampling(SDS) loss [24], leveraging priors from the diffusion model to guide the optimization and enhance fine-grained details in the resulting 3D vector graphic representation. This two-stage approach ensures both structural accuracy and high-fidelity detail preservation.

### 3.2 Stage I: Primary Structure Fitting

In the first stage, the model extracts key structural information from the reconstructed mesh $\mathcal{M}$ and uses it to fit 3D Bézier curves. The fitting process is further optimized using a specially designed Chamfer Distance loss.

#### 3.2.1 Salient Point Cloud Extraction

To identify high-curvature regions on the mesh, we adopt the Sharp Edge Sampling (SES) process from Dora [4] to extract a salient point cloud. We traverse each edge of the mesh. For each edge, if it belongs to two adjacent faces, we compute the angle between the normal vectors of these two faces. If the angle is below a predefined threshold, we consider this edge as a salient edge. To address the challenge of extracting salient edges from smooth surfaces (e.g., spheres), we uniformly sample camera parameters on a horizontal plane. For each sampled viewpoint, we compute the front faces and back faces. Edges shared by a front face and a back face are identified as silhouette edges. All such silhouette edges are subsequently incorporated into the salient edge set. After identifying all salient edges, we sample points along these edges to create a point cloud $\mathcal{P}_s$ as ground truth.

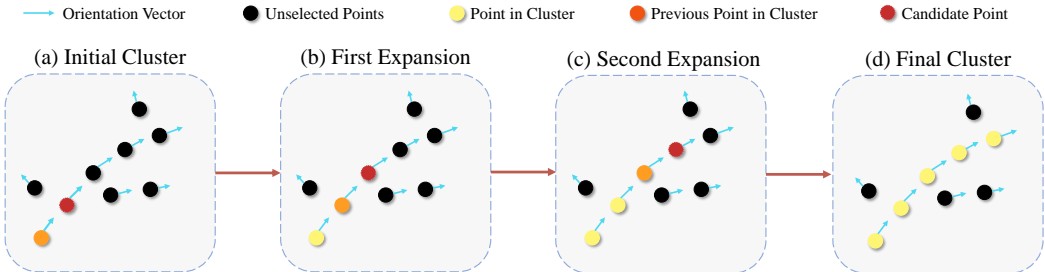

Figure 4: Visualization of point cloud clustering. Each point is assigned an orientation vector (blue arrows). In (a), the orange point initializes the cluster, and a candidate (red) is evaluated based on spatial proximity and orientation similarity. In (b), the candidate meets both criteria and is incorporated into the cluster. This process iterates until the final cluster is formed, as shown in (d).

### 3.2.2 Point Cloud Clustering

After obtaining the salient point cloud $\mathcal{P}_s$, we aggregate these discrete points into clusters suitable for Bézier curve fitting. Inspired by EdgeGaussians [3], we perform clustering for edge fitting based on vertex orientations. While EdgeGaussians directly utilize the principal directions of 3D Gaussians as orientation vectors, such directional information is absent in our discrete point cloud $\mathcal{P}_s$. To address this, we introduce an initialization step to estimate orientation vectors for each point in $\mathcal{P}_s$. Specifically, for each point $p$, we first identify its $k$ nearest neighbors and then apply Principal Component Analysis (PCA) [22] to the local neighborhood. The resulting primary eigenvector is used as an approximation of $p$'s orientation vector $\vec{v_p}$.

Once orientation vectors are assigned to each point, we partition the point cloud into multiple clusters using an iterative expansion process as shown in Fig 4. Each cluster is initialized from a randomly selected starting point and grows by progressively incorporating neighboring points that satisfy both spatial proximity and orientation similarity. Specifically, we use the most recently added point in the cluster, denoted as $p$, to guide the selection of the next candidate. A candidate point $q$ is added to the current cluster if it meets two criteria: (1) it lies within the spatial neighborhood of the cluster, i.e., $dis(p,q) \leq d_{\text{thresh}}$; and (2) the direction of the new edge formed by $p$ and $q$ aligns with the orientation vector of $q$, i.e., $\arccos\left(\left|\frac{\vec{pq}}{|\vec{pq}|} \cdot \frac{\vec{v_q}}{|\vec{v_q}|}\right|\right) < \theta_{\text{thresh}}$. The expansion continues until no additional points can be incorporated under these constraints, completing one cluster.

This process repeats across the point cloud to generate a complete set of directional clusters, each representing potential curves for subsequent Bézier fitting. The randomized clustering method ensures comprehensive coverage of all geometric features while avoiding bias toward specific regions. Finally, we filter the small clusters with fewer than $\tau$ points, as they are likely to be noise or outliers.

### 3.2.3 Bézier Curves Fitting

After obtaining these clusters, we attempt to fit them with either straight lines or Bézier curves, selecting the one with the least error as the fitting result for that cluster.

The fitting process is performed using the Chamfer Distance loss function [26] to minimize the distance between the Bézier curves and the salient point cloud $\mathcal{P}s$ obtained in Sec. 3.2.1. The fitted Bézier curves are denoted as $\{C_i\}_{i=1}^n$. For our implementation, we use cubic Bézier curves with four control points $P_0, P_1, P_2, P_3 \in \mathbb{R}^3$. The parametric equation for a 3D cubic Bézier curve can be written in explicit representation as:

$$B(t) = (1-t)^3 P_0 + 3(1-t)^2 t P_1 + 3(1-t)t^2 P_2 + t^3 P_3, \quad t \in [0,1] \tag{1}$$

To generate a point cloud from a set of Bézier curves $\{C_i\}_{i=1}^n$, we uniformly sample $s$ points along each curve by evaluating the parametric function $B(t)$ at $t_j = \frac{j-1}{s-1}$ for $j = 1, \ldots, s$. The resulting point cloud $\mathcal{P}_c$ is defined as:

$$\mathcal{P}_c = \big\{ B_i(t_j) \mid i \in \{1, \ldots, n\}, \, j \in \{1, \ldots, s\} \big\}. \tag{2}$$

Chamfer Distance loss is computed as follows:

$$\mathcal{L}_c(\tilde{\mathcal{P}}_c, P_s) = \frac{\lambda}{|\tilde{\mathcal{P}}_c|} \sum_{p \in \tilde{\mathcal{P}}_c} \min_{q \in \mathcal{P}_s} \|p - q\|^2 + \frac{1}{|\mathcal{P}_s|} \sum_{q \in \mathcal{P}_s} \min_{p \in \tilde{\mathcal{P}}_c} \|p - q\|^2 \tag{3}$$

where $\tilde{\mathcal{P}}_c$ denotes $\mathcal{P}_c$ augmented with Gaussian noise (introduced for data augmentation), and $\lambda$ is a hyperparameter to balance the two terms. The generation of point cloud $\mathcal{P}_c$, which relies on the Bézier curve formulation in Eq. 1, is differentiable with respect to the curve's control points. Consequently, the chain rule enables the gradients from $\mathcal{L}_c$ to propagate back to these control points, facilitating their iterative optimization.

### 3.3 Stage II: Detail Refinement

Some objects may contain intricate-to-approximate regions that Stage I might miss due to limitations in the salient point cloud extraction or clustering process. These regions are identified by analyzing the mesh's vertex distribution and locating areas not adequately covered by the point cloud $\mathcal{P}_c$ generated in Stage I. For such cases, we introduce an additional refinement stage to handle these regions by distilling priors from a pretrained image-to-3D model.

First, the parameters of the initial curves $\{C_i\}_{i=1}^n$ from Stage I are frozen. We randomly initialize new Bézier curves $\{C_i'\}_{i=1}^{n'}$ (with parameters $\theta'$) in regions that are intricate to approximate, thereby complementing the primary structure. To jointly represent both curve sets, we sample a combined point cloud $\mathcal{P}_{combined} = \mathcal{P}_c \cup \mathcal{P}_{c'}$ and encode it into a latent space $\mathcal{Z}$ using a pretrained VAE encoder from [15]: $z = \mathcal{E}(\mathcal{P}_{combined})$. Then we refine only the new parameters $\theta'$ via SDS loss [24], supervised by the input image $I$.

$$\nabla_{\theta'} \mathcal{L}_{\text{SDS}} = \mathbb{E}_{t,\epsilon} \left[ w(t) \left( \epsilon_\phi(z_t, I; t) - \epsilon \right) \frac{\partial z}{\partial \theta'} \right] \tag{4}$$

where $z_t$ is the noised latent variable at timestep $t$, $\epsilon_\phi$ is the denoising model conditioned on $I$, and $w(t)$ is a weighting function. This process iteratively adjusts the newly added Bézier curves to fill in missing details from Stage I, ensuring consistency and high fidelity in the final 3D vector representation $\mathcal{S}^{3D} = \{C_i\}_{i=1}^n \cup \{C_i'\}_{i=1}^{n'}$.

### 3.4 3D Vector Graphics Rendering

To enable both qualitative visualization and quantitative evaluation, 3D vector graphic $S^{3D}$ should be projected onto a 2D plane. As proved by 3Doodle [5], the perspective projection of a 3D Bézier curve onto a 2D plane yields a 2D rational Bézier curve. Given a 3D curve $B(t)$ and image plane at $z = f$ (where $z$ is the depth axis and $f$ the focal length), the projection is:

$$B^{2D}(t) = \begin{pmatrix} B_x(t) \frac{f}{B_z(t)} \\ B_y(t) \frac{f}{B_z(t)} \end{pmatrix} \tag{5}$$

Consequently, by explicitly defining the camera parameters in 3D space, we obtain a consistent set of 2D rational Bézier curves that correspond directly to the camera's viewpoint. These curves can then be faithfully rendered using DiffVG [14] to generate the corresponding SVG files, which are subsequently utilized for both comprehensive quantitative metrics and detailed qualitative analysis.

# 4 Experiment

## 4.1 Implementation Details

Our VC3D framework is implemented in PyTorch. For primary structure fitting stage, we set the distance threshold $d_{\text{thresh}} = 0.05$ and angle threshold $\theta_{\text{thresh}} = 50°$. Each Bézier curve is defined by 4 control points, with $s = 64$ sample points per curve for optimization. For detail refinement stage, we employ TripoSG [15] as our pretrained image-to-3D model, with SDS loss weight set to $2 \times 10^{-4}$. We use the SGD optimizer [29] with a learning rate of $5 \times 10^{-3}$.

All experiments are conducted on a single NVIDIA RTX 4090 GPU. For each input, our method typically produces fewer than 100 Bézier curves. Our full method takes about 30 minutes to generate a vector graphic, with 100 optimization steps for Stage I and 200 steps for Stage II. We collected 40 images from prior works and online sources as inputs. All generated 3D vector graphics are rendered into 12 views using identical camera parameters, upon which the metrics are computed.

## 4.2 Experimental Results

We compare our approach with two state-of-the-art methods in 3D vector graphics generation: Diff3DS [58], which designs a depth-aware differentiable rasterizer and leverages 2D diffusion model priors through SDS loss to generate 3D vector graphics from text or images, and 3Doodle [5], which employs perceptual losses with multi-view guidance to obtain 3D Bézier curve representations of objects. To comprehensively evaluate the quality and fidelity of the generated 3D vector graphics, we employ CLIPScore [28] to measure semantic alignment between rendered views and input images Furthermore, we use an aesthetic indicator [31] to quantify the aesthetic value.

### 4.2.1 Qualitative Evaluation

Figure 5 presents qualitative comparisons between our method and previous work, 3Doodle [5] and Diff3DS [58]. As shown, VC3D produces cleaner, more accurate, and more view-consistent 3D vector graphics. Previous works struggle to capture fine details in reference images, such as the patterns on butterfly or the handle of the coffee cup. Additionally, their outputs often contain excessive messy lines (e.g., the chair example). Figure 6 shows the rendering results of our generated results from different camera viewpoints.

### 4.2.2 Quantitative Evaluation

Table 1 presents the quantitative analysis results of all methods. Our method outperforms previous approaches in CLIPScore and Aesthetic Score. Our method achieves a cosine similarity of $0.799$, which is higher than the $0.729$ achieved by 3Doodle and the $0.673$ achieved by Diff3DS. At the same time, we achieved the highest score in Aesthetic Score. These superior results demonstrate our method's ability to generate semantically and geometrically superior 3D vector graphics.

In addition to the metrics mentioned above, our method demonstrates significant advantages in generation time. Our method requires only a few SDS loss optimization steps, significantly reducing generation time. The total runtime for two stages is approximately $0.5$ hours, showing notable improvements compared to 3Doodle (~6 hours) and Diff3DS (~2 hours).

Table 1: Quantitative comparison of VC3D and previous methods on evaluation metrics. The **bold numbers** represent the best performance.

| Method | CLIPScore ↑ | Aesthetic Score ↑ |
|---|---|---|
| 3Doodle | 0.729 | 4.122 |
| Diff3DS | 0.673 | 3.769 |
| VC3D (Ours) | **0.799** | **4.352** |

Table 2: Ablation study results comparing different variants of our proposed method. The **bold numbers** represent the best performance.

| Method | CLIPScore ↑ | Aesthetic Score ↑ |
|---|---|---|
| Variant 1 | 0.779 | 4.096 |
| Variant 2 | 0.805 | 4.167 |
| Full Method | **0.818** | **4.297** |

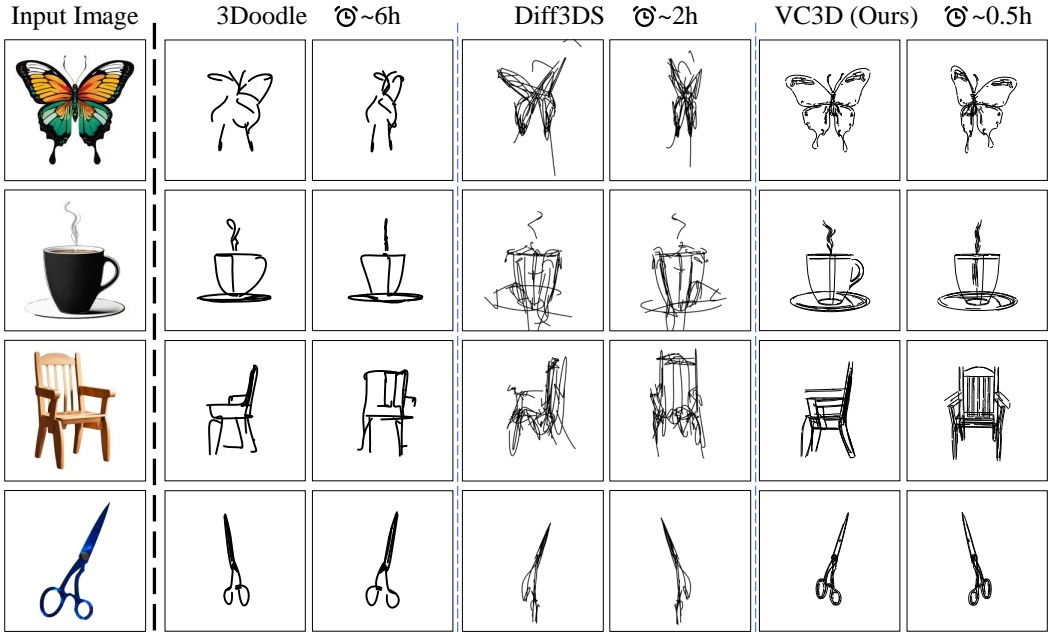

Figure 5: Qualitative comparison of different methods. Diff3DS and VC3D use a single image $I$ as input, while 3Doodle uses 120 rendered images of the mesh reconstruction result $\mathcal{M}$ as input.

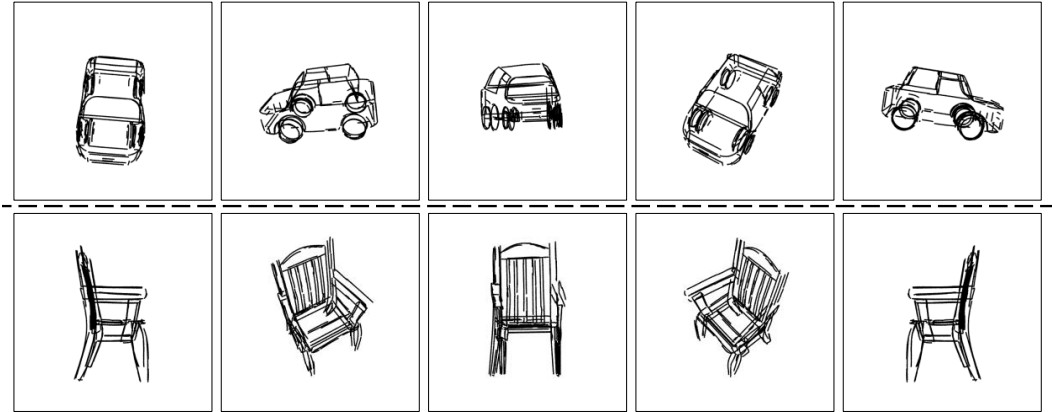

Figure 6: Multi-view results generated by our method VC3D.

### 4.3 Ablation Studies and Analysis

To demonstrate the respective contributions of the Chamfer Distance loss and SDS loss, we performed ablation experiments. We selected a subset of 20 images from the inputs in Section 4.2.1, where all examples were optimized with SDS loss. And results were recorded after three distinct stages, corresponding to three variants: (1) **Variant 1:** This variant refers to the model with the salient point cloud extraction and the point cloud clustering, (2) **Variant 2:** This variant is the model with the first stage only, *i.e.*, Primary Structure Fitting, and (3) **Full Method**: This is our proposed method, which comprises two stages. The results are shown in Table 2. The improvements of the **Variant 2** against the **Variant 1** indicate the benefits brought by using CD loss for optimization. Comparing the performance of **Variant 2** and **Full Method**, it is clear that the detail refinement stage can further improve the performance in CLIPScore and Aesthetic Score metrics.

In Figure 7, we show the optimization process of Chamfer Distance loss. The initially fitted Bézier curves often fail to accurately cover the salient point cloud $\mathcal{P}_s$. The coherence between curves is

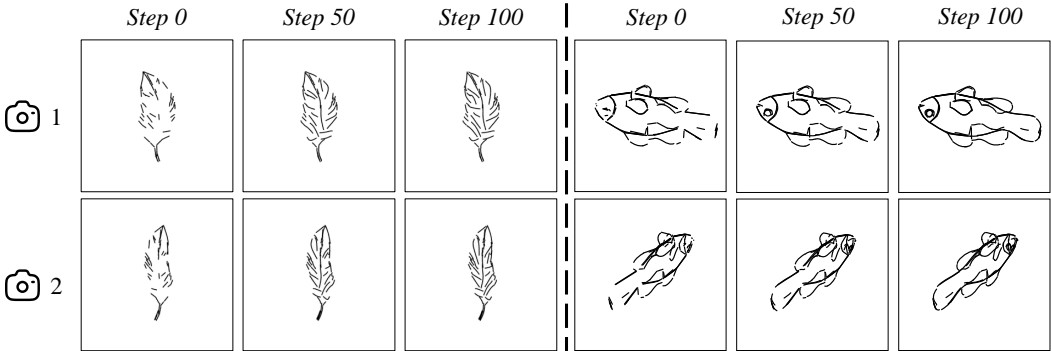

Figure 7: Illustration of the optimization process using Chamfer Distance loss $\mathcal{L}_c$.

Table 3: Ablation study results of only SDS loss. The **bold numbers** represent the best performance.

| Method | SDS-only | Full Method |
|---|---|---|
| CLIP score ↑ | 0.622 | **0.799** |
| Aesthetic Score ↑ | 3.653 | **4.352** |

also suboptimal. As optimization progresses, the curves gradually extend to form more complete structures, ultimately achieving both improved coverage of the salient features and enhanced inter-curve coherence while preserving the overall geometric fidelity of the original shape.

Primary Structure Fitting (Stage I) is a critical component that significantly enhances both generation quality and efficiency. To demonstrate its importance, we conduct an ablation study by removing Primary Structure Fitting and solely relying on SDS loss for optimization. Results are shown in Table 3. It can be observed that the CLIPScore decreases from 0.799 to 0.622, and the Aesthetic Score drops from 4.352 to 3.653. These reductions signify a marked deterioration in the visual quality of the generated outputs. Moreover, the SDS-only optimization is extremely inefficient, requiring approximately 3 hours to generate a vector graphic. Similar results are also observed in related works like DiffSketcher [45] (Figure 8). Therefore, Primary Structure Fitting is an indispensable step that provides the necessary structural foundation for the refinement stage.

The visual improvements brought by the SDS loss can be observed in Figure 8, where the refinement stage compensates for previously overlooked details (e.g., terminal branches on corals) and improves the structural coherence of the 3D vector graphics.

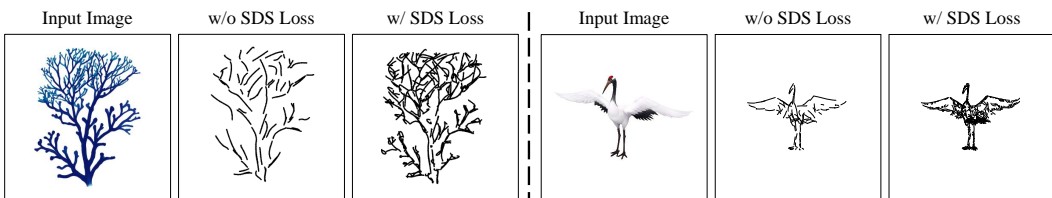

Figure 8: Illustration of the optimization effect of SDS loss $\mathcal{L}_{sds}$. SDS loss effectively recovers missing structural information while enhancing geometric detail representation.

These results demonstrate that our two-stage approach effectively balances structural accuracy with visual quality, leading to more compelling and semantically accurate 3D vector graphics.

We also experimented with the number of Bézier curves. Since the number of Bézier curves is equal to the number of point clusters, we can control the number of curves by adjusting the cluster count, i.e., by changing the filtering threshold $\tau$ in Point Cloud Clustering stage. As shown in Figure 9, when $\tau = 10$, point clusters with fewer than 10 points are removed. Increasing the threshold eliminates more clusters, reducing the number of Bézier curves retained and producing an abstract result. We conducted an experiment using ten relatively complex examples selected from those

used in quantitative evaluation. The quantitative results are presented in Table 4. We can see with the filtering threshold $\tau$ increased, the CLIP score and Aesthetic score decrease, suggesting that geometric details are gradually reduced.

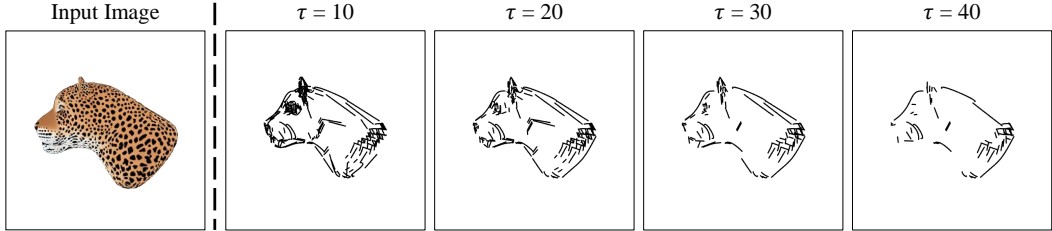

Figure 9: Effect of the filtering threshold in Point Cloud Clustering. The number of Bézier curves gradually decreases as $\tau$ increases, producing a more abstract effect.

Table 4: Ablation study results of different $\tau$. The **bold numbers** represent the best performance.

| $\tau$ | **5** | **10** | **20** | **30** | **40** |
|---|---|---|---|---|---|
| CLIP score ↑ | 0.7523 | **0.7676** | 0.7544 | 0.7340 | 0.7214 |
| Aesthetic Score ↑ | **4.3918** | 4.3567 | 4.2349 | 4.1393 | 4.0580 |

## 5 Limitations and Future Works

While VC3D efficiently generates view-consistent 3D vector graphics, it is noted that individual curves may inadvertently traverse distinct semantic boundaries. Future work could address this limitation by incorporating 3D segmentation results as a supervisory signal. This approach would refine the curve initialization process, ensuring the generated curves align more accurately with the underlying part structures of the object.

In addition, considering that our method can generate corresponding 3D vector graphics from meshes with minimal time cost, we can build 3D vector graphics datasets based on open-source mesh datasets in the future, providing a research foundation for subsequent work.

## 6 Conclusion

In this paper, we present VC3D, a novel framework for generating view-consistent 3D vector graphics using 3D priors. Operating directly in 3D space rather than 2D projection planes, our approach effectively addresses view inconsistency issues. Our two-stage algorithm first identifies salient structures through geometric clustering and Bézier curve fitting, then refines results using SDS loss with a pretrained image-to-3D model. Experiments demonstrate that VC3D preserves geometric characteristics while maintaining view consistency across viewpoints, with advantages in generation efficiency. This research makes high-quality 3D vector graphics creation more accessible and applicable to virtual reality, shape retrieval, and conceptual design.

## 7 Acknowledgments

This work was supported in part by the National Key Research and Development Project of China (No. 2022ZD0117801) and Young Elite Scientists Sponsorship Program by CAST, in part by National Natural Science Foundation of China (No.62572039, No.62461160331, No.62132001), in part by Zhejiang Provincial Natural Science Foundation of China under Grant (No. LQN26F020041), in part by the Fundamental Research Funds for the Central Universities. This work was also supported by the NSFC/RGC Collaborative Research Scheme (CRS_HKU703/24). Dr. Xu's research work described in this paper was conducted in the JC STEM Lab of Multimedia and Machine Learning funded by The Hong Kong Jockey Club Charities Trust.

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

## Technical Appendices and Supplementary Material

## Overview

This supplementary material is organized into several sections that provide additional details and analysis related to our work on ViewCraft3D (VC3D). Specifically, it includes the following aspects:

- In Section A, we provide detailed implementation information for VC3D.
- In Section B, we present additional qualitative results generated by VC3D.
- In Section C, We compared VC3D with the concurrent work Dream3DVG.
- In Section D, we describe a user study that demonstrates the superiority of our method compared to existing approaches.
- In Section E, we discuss the applications of VC3D and 3D SVGs.
- In Section F, we discuss the potential societal impact of VC3D.

## A    Implementation Details of VC3D

### A.1    Algorithm Flow of Salient Point Cloud Extraction

Algorithm 1 shows the details of Salient Point Cloud Extraction (in main paper Sec. 3.2.1).

---

**Algorithm 1:** Salient Point Cloud Extraction

---

**Input:** Mesh file $\mathcal{M}$, Boolean variable $enable\_find\_silhouette\_edges$
**Output:** Salient point cloud $\mathcal{P}_s$
$\mathcal{P}_s \leftarrow \emptyset$ ;                            // Initialize the salient point cloud
$\mathcal{P}_s \leftarrow SES\_Process(\mathcal{M})$ ;             // Extract salient points using SES
**if** $enable\_find\_silhouette\_edges$ **then**
  $E \leftarrow Find\_Silhouette\_Edges(\mathcal{M})$ ;          // Extract silhouette edges
  $\mathcal{P}_e \leftarrow Sample\_Points(E)$ ;        // Sample points on silhouette edges
  $\mathcal{P}_s \leftarrow \mathcal{P}_s \bigcup \mathcal{P}_e$ ;                        // Merge point cloud
**return** $\mathcal{P}_s$ ;                      // Return the salient point cloud

---

SES stands for Sharp Edge Sampling, a technique proposed in Dora [4] for extracting salient point clouds. The parameter $enable\_find\_silhouette\_edges$ is a hyper-parameter that controls whether silhouette edges are detected. We set this parameter to True when the mesh contains many smooth surfaces (e.g., spheres, cylinders). Figure 10 illustrates the results on the same example with $enable\_find\_silhouette\_edges$ set to both True and False for comparison.

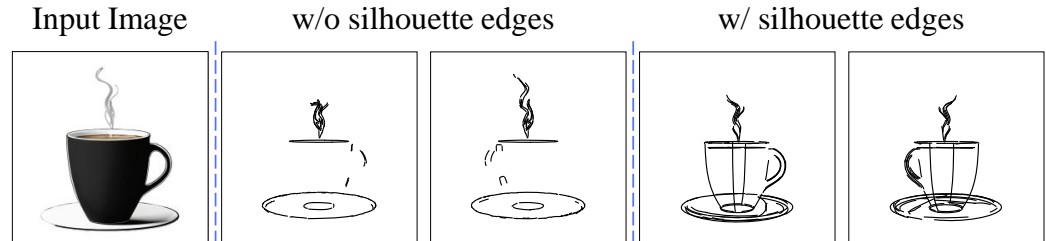

Figure 10: Effect of Silhouette Edge Extraction. Comparison of outputs with (right) and without (middle) the silhouette edge extraction enabled.

When the parameter $enable\_find\_silhouette\_edges$ is set to False, the salient edge extraction fails to capture the outline of the cup body, resulting in an incomplete representation. In contrast, enabling this parameter (*i.e.*, setting it to True) allows the extraction process to effectively highlight the cup body, which is clearly reflected in the final result.

## A.2 Details of Detail Refinement

In the Detail Refinement stage, we randomly add $N$ Bézier curves to intricate-to-approximate regions (as defined in Sec. 3.3 of main paper). By default, $N$ is set to 128, but it can be adjusted based on the geometric complexity of missing areas. The intricate-to-approximate regions are determined by the mesh vertices and the point cloud $\mathcal{P}_s$. Specifically, for a point $p$, if its distance to all points in the point cloud $\mathcal{P}_s$ exceeds a threshold (set to $0.03$ in our experiments), it is considered to belong to the intricate-to-approximate regions. All such points comprise a set $\mathcal{P}_i$, from which vertices are randomly selected to initialize the Bézier curves.

For both the Bézier curves from the first stage and those newly added in the second stage, we sample $64$ points per curve to construct the combined point cloud $\mathcal{P}_{combined}$. Based on the mesh $\mathcal{M}$ generated in the first stage, each point in $\mathcal{P}_{combined}$ is assigned an approximate normal direction as a geometric prior. Both $\mathcal{P}_{combined}$ and corresponding normal directions are then encoded using the VAE encoder of TripoSG [15] to generate structured latent representations. To ensure stable optimization, the SDS loss [25] employs medium-strength noise with the range $[0.15, 0.4]$ and multiple denoising steps. This strategy effectively balances global shape consistency with local detail preservation, thereby enhancing the visual quality of the generated vector graphics.

# B More Qualitative Results

## B.1 More Qualitative Results of VC3D

As shown in Figure 11, VC3D is capable of generating high-quality results across a diverse range of 3D object categories, such as animals, industrial products, and furniture.

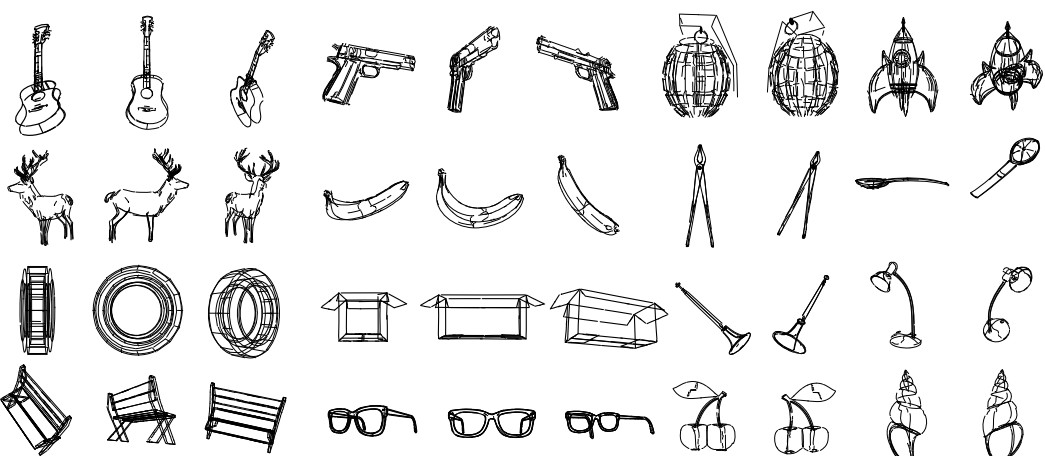

Figure 11: More qualitative results generated by our method VC3D.

## B.2 More Comparisons with Other Methods

Figure 12 presents additional comparisons between our method and baseline methods. It can be observed that VC3D outperforms these methods in terms of fidelity and view-consistency.

| Input Image | 3Doodle | Diff3DS | VC3D (Ours) |
|:-:|:-:|:-:|:-:|

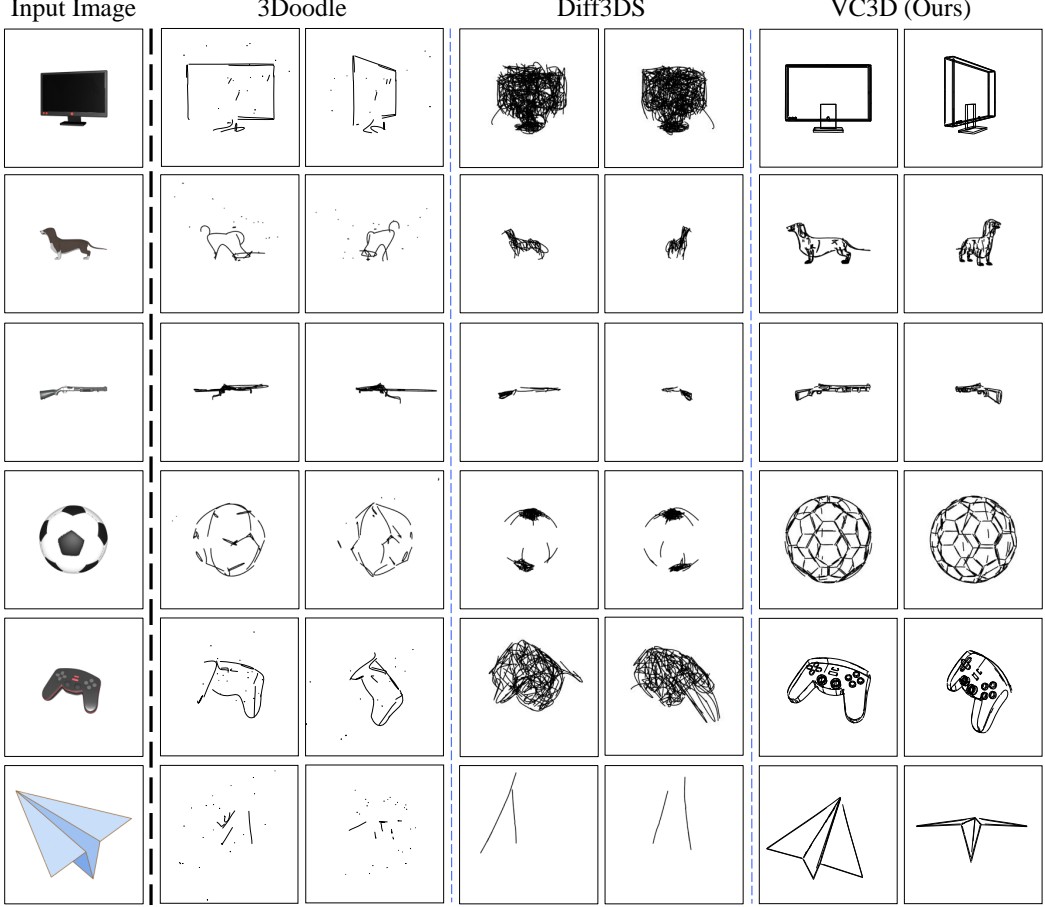

Figure 12: More qualitative comparisons between our method and baseline methods, 3Doodle and Diff3DS.

## C    Comparison with Dream3DVG

We conducted a quantitative comparison with Dream3DVG [52]. Specifically, given that Dream3DVG accepts text prompts as input, we manually convert the input images from our quantitative experiments into text prompts and use them as input for Dream3DVG. As shown in Table 5, our method achieves a higher aesthetic score than Dream3DVG.

Table 5: Quantitative comparison of VC3D and Dream3DVG [52]. The **bold numbers** represent the best performance.

| Method | VC3D (Ours) | Dream3DVG |
|:-:|:-:|:-:|
| Aesthetic Score ↑ | **4.352** | 4.1499 |

## D    User Study

We conducted a user study to evaluate the quality of the synthesized 3D vector graphics. The test set used in the study is identical to that in the main paper and consists of 40 cases, each with results generated by three different approaches. For the user study, each questionnaire contained 20 cases randomly sampled from these 40.

We invited 30 volunteers to participate in the study, with a gender ratio of approximately 5:1 (male to female). Most participants are university students from various science and engineering disciplines, with ages ranging from 18 to 27. Volunteers were asked to choose the best result for each case based on two criteria: fidelity and view-consistency. As shown in Fig. 13, our approach received 71.17% of the votes, outperforming 3Doodle (13.83%) and Diff3DS (15%).

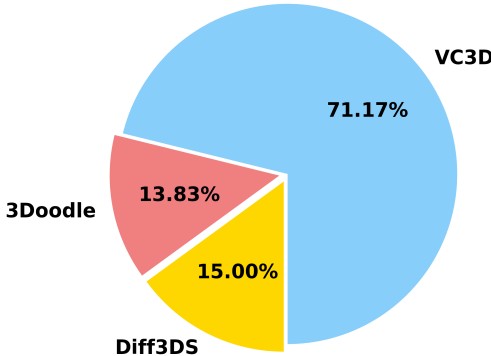

Figure 13: The results of the user study.

# E    Applications of VC3D and Resultant 3D SVGs

The inherent editability of 3D vector graphics offers significant advantages for iterative modeling workflows, a potential explored in several academic works. (1) 3D vector curves can serve as real-time guidance for surface creation and editing. For instance, FiberMesh [21] enables the design of freeform surfaces by allowing users to directly draw and manipulate a network of 3D curves, which function as a structural vector scaffold for the surface. (2) Some works focus on reconstructing 3D surfaces from a network of 3D vector curves, treating the curve network as an editable and interpretable representation that facilitates intuitive shape refinement. Notably, True2Form [48] infers a complete 3D curve network from ambiguous 2D sketches, bridging the gap between sparse 2D input and structured 3D geometry.

3D vector graphics also support animation production. For example, Blender provides an official SVG plugin and keyframe interpolation tools. Users can project 3D vector models onto different viewpoints, and then generate animations using the plugin. This enables the creation of 2D animations from multiple angles while maintaining strong frame-to-frame consistency.

3D vector graphics are integral to VR/AR applications, and they have already been applied in some well-known software, such as Tilt Brush, Gravity Sketch, and Mental Canvas. With the rapid development of VR/AR devices and applications, 3D vector graphics will become more valuable. Our proposed approach can be used to generate 3D vector graphics assets.

# F    Societal Impact

Our method, VC3D, significantly enhances the fidelity and view-consistency of 3D vector graphics generated from a single image. However, as with other generative models, there exists a critical concern regarding the potential negative societal impact if our model is misused to create inauthentic or misleading 3D vector content. It is imperative to ensure that the model is used responsibly to mitigate the risk of adverse social consequences.

