# OpenReview forum: "ViewCraft3D: High-fidelity and View-Consistent 3D Vector Graphics Synthesis"
_NeurIPS.cc/2025/Conference — NeurIPS 2025 poster_

### Official Review · Reviewer_221W · 2025-06-22

**Clarity:** 2
**Significance:** 3
**Originality:** 2
**Rating:** 4
**Confidence:** 4

**Summary:**

The paper proposes an efficient pipeline for generating 3D vector graphics from a single image. The method first reconstructs a 3D mesh using an initial image-to-3D model, then fits vector graphics to this mesh, and finally leverages a pretrained model to refine the details, resulting in view-consistent and high-quality output.

**Questions:**

My main concerns relate to missing details and a lack of complexity analysis. I would reconsider my rating if the authors could clarify:
 Bézier Fitting: What determines the number of curves to fit to a point cloud? How are regions for refinement identified, and how are new curves initialized?
 Complexity: Your method seems to produce more curves than baselines (e.g., 3Doodle). Can you provide a quantitative comparison and discuss this complexity-quality trade-off?
 Diffusion Model: Can you provide more details on the architecture and operation of the diffusion model used for refinement?
Addressing these points would significantly strengthen the paper's technical contribution and clarity.

**Ethical Concerns:**

["NO or VERY MINOR ethics concerns only"]

**Final Justification:**

The rebuttal has addressed most of my concerns.

**Limitations:**

Yes,

**Paper Formatting Concerns:**

None.

**Quality:**

2

**Strengths And Weaknesses:**

The authors have developed a functional and well-structured pipeline that successfully addresses the task of image-to-3D-vector-graphics conversion. The strength of the work is its clear demonstration of how to chain existing models to achieve a desirable result.

However, the paper's main drawback is its reliance on pre-existing components. The pipeline's architecture is a straightforward combination of established techniques, and the core generative model is borrowed from another source. Consequently, the paper's fundamental technical contribution is not as significant as one might hope for.

---

> ### Author Rebuttal · Authors · 2025-07-31
>
> ### **W1: Novelty.**
>
> While our pipeline leverages pretrained models, our core contribution is the introduction of a **novel framework that directly addresses the fundamental issue of view-inconsistency** prevalent in existing 3D graphics generation methods. Previous works, such as 3Doodle[1], Diff3DS[2], and Dream3DVG[3], rely heavily on **2D constraints**: they project 3D models into 2D views and utilize differentiable rendering to enforce alignment between the projections and image ground-truth. However, this reliance on 2D priors inherently limits their ability to enforce true 3D coherence across multiple views.
>
> In contrast, **our method introduces a paradigm shift by incorporating 3D priors directly into the optimization process for 3D vector graphics**. Specifically, we extract salient edges from the mesh and leverage a pretrained 3D generative model to define a 3D SDS loss in latent space, thereby completely removing the dependency on 2D projections. This enables optimization to be guided by a **holistic understanding of 3D shape geometry**, rather than by enforcing consistency across 2D views.
>
> To the best of our knowledge, **ViewCraft3D is the first framework to apply 3D SDS loss for optimizing structured 3D vector graphics**, which introduces both conceptual and technical novelty beyond a simple combination of existing components.
>
> ### **Q1: Bézier Fitting.**
>
> **Number of Curves:** The number of curves is jointly determined by the number of clusters, resulting from the Point Cloud Clustering stage (Section 3.2.2), and the filtering threshold, $\tau$. The filtering threshold removes clusters with fewer than $\tau$ points, and each of the remaining clusters then has a Bézier curve fitted to it.
>
> **Region Identification and Curve Initialization:** As we detailed in Lines 204-214, regions for refinement are identified by analyzing the mesh's vertex  distribution, locating areas not adequately covered by the point cloud $\mathcal{P}_c$  from Stage I (Section 3.2). Specifically, a vertex is considered  "uncovered" if there are no points from point cloud $\mathcal{P}_c$ within a radius $R$ of that vertex. To initialize new Bézier curves, we first apply K-Means clustering to uniformly sample points from the uncovered point set, using them as initial control points. The remaining control points are then randomly selected within a confined local region around each initial point, forming the initial curves.
>
> ### **Q2: Complexity.**
>
> Thank you for the question. Our method may produce more curves. **However, this reflects the model’s ability to capture finer geometric details**. To quantitatively investigate the complexity–quality trade-off, we conducted an experiment using ten relatively complex examples selected from those reported in the main paper. The quantitative results are presented in the table below.
>
> |  $\tau$               | 10      | 20      | 30      | 40     |
> |------------------|---------|---------|---------|--------|
> | CLIP score↑      | 0.7676  | 0.7544  | 0.7340  | 0.7214 |
> | Aesthetic Score↑ | 4.3567  | 4.2349  | 4.1393  | 4.0580 |
>
>
> We can see with the filtering threshold τ  increased, the CLIP score and Aesthetic score decrease, suggesting that geometric details are gradually reduced.
>
> ### **Q3: Details of Diffusion Model.**
>
> The goal of the refinement stage is to improve the initial Bézier curve representation obtained in Stage I by leveraging 3D generative priors via **Score Distillation Sampling (SDS)**. SDS aims to generate a pseudo-supervisory signal from a pretrained diffusion model (in our case, an image-conditioned 3D diffusion model) and iteratively adjusts the target (here, the Bézier curve parameters $\theta$) via gradient descent.
>
> Our method builds upon the architecture of TripoSG, which comprises two main components: **3D Variational Autoencoder (VAE) and a Rectified Flow Transformer**. In our pipeline, we first sample a point cloud from the Bézier curves and encode it into the latent space $\mathcal{Z}$ using the pretrained VAE encoder from TripoSG, obtaining a latent code $z$.
>
> We then apply SDS by perturbing this latent code and feeding it into the Rectified Flow Transformer, which returns a score (i.e., a denoising direction). This score acts as a gradient signal that suggests how the latent code—and consequently the underlying 3D curves—should be modified to better match the priors captured by the diffusion model. We backpropagate this gradient through the VAE encoder and the point sampling process to update the Bézier curve parameters $\theta$, thereby refining the 3D curves in a differentiable, geometry-aware fashion.
>
> [1] 3Doodle: Compact Abstraction of Objects with 3D Strokes
>
> [2] Diff3DS: Generating View-Consistent 3D Sketch via Differentiable Curve Rendering
>
> [3] Empowering Vector Graphics with Consistently Arbitrary Viewing and View-dependent Visibility

---

> > ### Comment · Reviewer_221W · 2025-08-05
> >
> > The rebuttal has addressed most of my concerns. And I have raised my rating.

---

> > > ### Author Response · Authors · 2025-08-05
> > >
> > > Dear Reviewer 221W,
> > >
> > > Thank you for the opportunity to provide additional clarifications. We also sincerely appreciate your positive feedback and the thoughtful reconsideration of our paper.

---

> ### Author Response · Authors · 2025-08-05
> **Welcome to discuss**
>
> Dear Reviewer 221W,
>
> Thank you again for your valuable feedback on our paper.
>
> We have submitted our rebuttal, which addresses the concerns you raised, including our novelty, algorithm details and complexity–quality trade-off.
>
> As the discussion period concludes in two days, we would be grateful if you could take a moment to review our response and let us know if you have any further questions.

---

### Official Review · Reviewer_vumt · 2025-06-30

**Clarity:** 3
**Significance:** 2
**Originality:** 3
**Rating:** 4
**Confidence:** 3

**Summary:**

The paper introduces ViewCraft3D (VC3D), a multi-stage method for generating high-fidelity, view-consistent 3D vector graphics by incorporating 3D priors directly in 3D space. First, it extracts salient edges from meshes to create a point cloud, which is clustered using KNN and PCA for Bézier curve fitting. Second, it refines details by adding and optimizing additional Bézier curves with a Score Distillation Sampling (SDS) loss guided by a pretrained image-to-3D model. Experiments show that VC3D surpasses prior methods in CLIPScore, aesthetic quality, and runtime, achieving better view consistency and efficiency.

**Questions:**

1. I would appreciate a more concrete example to show how the generated 3D vector graphics could benefit sub-stream application.
2. In the limitations, the authors mention that they cannot handle occlusion relationship. It would be helpful to include qualitative examples demonstrating these failure cases to better understand the scope of this limitation.

**Ethical Concerns:**

["NO or VERY MINOR ethics concerns only"]

**Final Justification:**

I would suggest that the author provide more examples to illustrate the practicality. Overall, I intend to recommend acceptance of this paper.

**Limitations:**

Yes.

**Paper Formatting Concerns:**

N.A.

**Quality:**

2

**Strengths And Weaknesses:**

**Strengths**
1. The method is clearly presented and demonstrates improvement in performance over competitors on quantitative metrics.

2. Ablation studies validate the contributions of both the Chamfer loss and the SDS loss.

3. The method leverage explicit 3D geometric priors for generating view-consistent 3D vector graphics.

**Weakness**
1. The most prominent improvement on 3D consistency appears dependent on the quality of the mesh obtained from the previous image-to-3D generative model. The subsequent steps just further refine the quality of generated 3D vector graphics.

2. The authors should clarify why generating a 3D vector graphics from a 3D mesh—already obtained via an image-to-3D generative model—offers a more meaningful or useful representation for the sub-stream tasks **than simply using the 3D mesh itself or the point clouds extracted from it**. My concern is the application scenario is a little bit narrow if there are only aesthetics values. As mentioned before, the proposed approach relies on meshes (already a concrete, ultimate 3D format) obtained by an additional 3D generative step to instead produce an intermediate representation (the 3D vector graphics), which seems somewhat counterintuitive as it makes the whole pipeline becomes a task-driven solution.

---

> ### Author Rebuttal · Authors · 2025-07-31
>
> ### **W1: The Role of Reconstructed Mesh.**
>
> Thank you for your question. The most critical idea of our approach is to achieve view consistency by leveraging 3D priors for 3D vector graphics generation. The 3D priors include geometric priors from the reconstructed mesh (Stage I) and generative priors from a pretrained 3D generative model (Stage II). Therefore, the quality of the generated 3D vector graphics is not simply determined by the quality of the mesh.
>
> Specifically, in Stage I, we extract only the most salient edges from the reconstructed mesh, intentionally filtering out noise and surface details to mitigate the influence of potential artifacts or inaccuracies in the mesh. In Stage II, the vector graphic is further refined via SDS optimization, using 3D generative priors learned from large-scale 3D assets.
>
> As a result, view consistency achieved by our method is not passively inherited from the mesh, but is actively enforced through our two-stage pipeline and the integration of 3D priors.
>
> ### **W2 & Q1: Application of 3D Vector Graphics.**
>
> We appreciate the reviewer’s concern regarding the utility of 3D vector graphics beyond aesthetic purposes. Our method produces a lightweight view-consistent 3D vector representation that offers distinct advantages for downstream tasks where **interpretability, interactivity, and performance efficiency** are critical.
>
> For example, in an **AR-based industrial assistance application**, the generated vector graphic can be overlaid on real-world machinery to provide intuitive visual guidance. Instead of rendering a full textured 3D mesh, which is often computationally expensive, the vector abstraction can highlight the outline of a component that needs replacement or trace the routing of a cable, enabling fast, responsive, and informative AR interactions, especially on resource-constrained devices.
>
> Beyond AR, our representations are also valuable in **CAD and industrial design**, where curve-based editing and structural manipulation are standard. Designers can interactively modify part outlines, reuse components, or perform topological edits without operating on dense, hard-to-parse mesh data. Similarly, in **education and training**, 3D vector graphics allow learners to better understand object structures through simplified line-based visualizations, which can emphasize functionally important parts over surface-level details.
>
> Thus, our use of 3D vector graphics is not merely for aesthetics, but reflects a **deliberate design trade-off that prioritizes editability, abstraction, and semantic clarity**. Such capabilities are not inherently provided by raw 3D meshes. For more specific application scenarios, please refer to our response to Reviewer ztRZ W3.
>
> ### **Q2: Occlusion Limitation.**
>
> Sorry for the confusion. The occlusion limitation mentioned in the paper specifically refers to the **2D rendering of the 3D vector graphic for visualization purposes**, and is not related to the quality of the generated 3D curve representation itself. Since our current rendering assumes the same opacity and no depth ordering (i.e., all curves are drawn with the same style regardless of their spatial relationships), this can occasionally lead to reduced visual clarity and spatial interpretability, especially in cases involving dense overlaps or self-occluding structures. We will include qualitative examples of failure cases in the revised version.

---

> ### Author Response · Authors · 2025-08-05
> **Welcome to discuss**
>
> We appreciate the reviewer's thoughtful engagement with our work. We believe the additional experiments and clarifications we have provided should address the concerns raised. We welcome any further questions or comments during the remaining discussion period and are committed to providing comprehensive responses.

---

> > ### Comment · Reviewer_vumt · 2025-08-05
> > **Reply to authors**
> >
> > Thank you for your further clarification. I will maintain my positive score. However, I recommend that the authors, if possible, provide more concrete examples to illustrate how the generated 3D vector graphics enhance the listed downstream tasks, in order to better demonstrate the practical value of the proposed method.

---

> > > ### Author Response · Authors · 2025-08-06
> > >
> > > Thank you again for your affirmation and support for our work! Regarding the concrete example of applications, for instance, in the application of animation, the superior editability of the 3D vector graphics allows users to easily manipulate key paths or parameters, enabling the rapid and efficient creation of smooth vector animations. We will incorporate more concrete examples in the next version of our paper to better demonstrate the practical value of our method.

---

### Official Review · Reviewer_X23k · 2025-07-01

**Clarity:** 3
**Significance:** 2
**Originality:** 2
**Rating:** 4
**Confidence:** 4

**Summary:**

This paper proposes ViewCraft3D, a pipeline for generating 3D parametric curves from a single image. VC3D first generates a 3D mesh using a pretrained image-to-3D generator. Then, salient points are extracted from the mesh, and heuristic point cloud clustering is applied to the sampled points to form clusters that constitute potential curves. Subsequently, the authors propose Bezier curve fitting to optimize the parameters of cubic Bezier curves, using the Chamfer distance between the original sampled points and resampled points from the curves as the optimization objective. SDS loss is also introduced as a refinement step to enhance detail. Quantitative experiments using CLIP score and aesthetic score as metrics demonstrate improvements over previous methods. VC3D also shows a significant advantage in optimization time.

**Questions:**

Questions:

1.	Whether and how can VC3D optimize concise yet globally informative long strokes? (Regarding Weakness 2)

2.	What would results using only SDS loss yield? Does SDS loss in Stage II contradict Phase I optimization? (Regarding Weaknesses 3 & 4, this is crucial for evaluating the contribution.)

3.	Could VC3D be adapted to other 3D generation frameworks other than Triposg?

**Ethical Concerns:**

["NO or VERY MINOR ethics concerns only"]

**Final Justification:**

The authors have addressed my concerns in the rebuttal, particularly by further elaborating on the feasibility of using TripoSG as SDS guidance, which demonstrates the soundness of the proposed approach. Therefore, I am inclined to believe that this work should be accepted.

**Limitations:**

Unable to reconstruct textures, and generated line segments appearing fragmented. Details refer to Weaknesses.

**Paper Formatting Concerns:**

No. The format is correct.

**Quality:**

3

**Strengths And Weaknesses:**

Strengths:

1.	This paper is well-written and easy to follow.

2.	Although the proposed method is straightforward, the numerical experiments demonstrate its effectiveness to some extent (but I still have some questions according to the provided qualitative results).

3.	The experiments are sufficiently comprehensive, with most tests effectively demonstrating the characteristics of different components introduced.

Weaknesses:

1.	Texture cannot be reconstructed, only shape can be recovered. For example, the leopard's skin spots in Figure 8 fail to be reconstructed. This is likely to be limited by the pretrained image-to-mesh generator [1].

2.	The Point Cloud Clustering proposed in Stage 1 is too simple, and seems cannot handle complex, global lines (longer strokes) in images. This is evident in Figure 7 and Figure 8 (e.g., τ = 40). Compared to Figure 1 in CLIPasso [2], which uses fewer but more conclusive strokes, Point Cloud Clustering produces fewer continuous lines to outline objects.

3.	Regarding Figure 7 (wo SDS), Primary Structure Fitting only generates initial structures, while SDS loss *mainly* ensures visual similarity between reconstructed 3D vectors and inputs. This may diminish the paper’s claimed contribution as it seems that SDS is more important. An ablation study (e.g., SDS-only results) should visually clarify Primary Structure Fitting’s role.

4.	Additionally, SDS employs $\epsilon_{\phi}(z_t, I; t)$, relying on a point cloud-based pre-trained distribution (spread across surfaces) [1]. Could this contradict the previously extracted salient points (which are probably the edge points)?

[1] Triposg: High-fidelity 3d shape synthesis using large-scale rectified flow models

[2] CLIPasso: Semantically-Aware Object Sketching

---

> ### Author Rebuttal · Authors · 2025-07-31
>
> ### **W1: Texture Synthesis.**
>
> Thank you for this insightful comment. The absence of texture is an intentional design choice, as our paper's core objective is to generate view-consistent 3D vector graphics by leveraging 3D geometric priors. Our method focuses exclusively on these geometric features to solve the central challenge of view consistency for an object's structure. We agree that incorporating texture is an important direction, and we will explore it in our future work.
>
> ### **W2 & Q1: Clustering Method and Ability of Producing Long Strokes.**
>
> Our clustering method is capable of generating long and continuous strokes, as demonstrated by the global outlines of the *coffee cup* and *scissors* in Figure 5. Unlike 2D abstraction approaches such as CLIPasso, which are optimized for perceptual minimalism, our task requires balancing coherence across multiple viewpoints with localized geometric detail. Therefore, there is a trade-off between stroke length/number and detail preservation.
>
> Specifically, for objects with rich details, lowering hyperparameter $\tau$ retains more clusters, yielding more Bézier curves and richer details. For objects best described by long strokes, increasing hyperparameter $d_{thresh}$ encourages each cluster to extend longer, thereby yielding longer Bézier curve fitting results.
>
> ### **W3 & Q2: The Role of Primary Structure Fitting.**
>
> Thank you for your question. Primary Structure Fitting is a critical component that significantly enhances both generation quality and efficiency. To demonstrate its importance, we conduct an ablation study by removing Primary Structure Fitting (Stage I) and relying solely on SDS loss for optimization. The results are shown in the following table:
>
> |                  | SDS-only | Full Method |
> |------------------|----------|-------------|
> | CLIPScore↑       | 0.622    | 0.799       |
> | Aesthetic Score↑ | 3.653    | 4.352       |
>
>
> We can see that the CLIPScore decreases from 0.799 to 0.622, and the Aesthetic Score drops from 4.352 to 3.653. Additionally, we perform qualitative comparison and observe a noticeable degradation in 3D view consistency compared to our full method. The visualization results show that the lines in SDS-only generated results are more cluttered and lack sufficient structural lines. Moreover, the SDS-only optimization is extremely inefficient, requiring approximately 3 hours to generate a vector graphic. Similar results are also observed in related works like DiffSketcher[1] (Figure 8). Therefore, Primary Structure Fitting is **an indispensable step that provides the necessary structural foundation for the refinement stage**. We will include this ablation study in the next version.
>
> ### **W4: Potential Contradict.**
>
> We appreciate the reviewer’s insightful observation regarding the potential contradiction between the point cloud prior used in SDS and the salient edge-based curve fitting in Stage I.
>
> As we expalined in Line209-211 of our submission, the final 3D SVG composite of two sets of Bezier curves, curves from Stage I and II. Specifically, curves from **Stage I** fit to salient edges, capturing the primary structural features of the mesh. At **Stage II**, we initialize new curves in regions that are geometrically complex but not well approximated by Stage I, such as flat or smooth surfaces with fine-scale details. This design ensures that the aggregated point cloud better aligns with the distribution expected by the pretrained model used in SDS, which is based on full-surface point clouds.
>
> Moreover, our empirical results show that the TripoSG encoder is still capable of encoding the Bézier curve-sampled point cloud into valid latents. By applying noise to the latent code and denoising through the Rectified Flow model, we can reconstruct the overall 3D shape effectively. This demonstrates that, although our point cloud is sampled differently, it still falls within the distribution that TripoSG can effectively encode.
>
> ### **Q3: Adapt to Other 3D Generation Frameworks.**
>
> Our VC3D can be adapted to other 3D generative frameworks beyond TripoSG. The key requirement is that the framework must **provide an encoder**. This encoder is essential for our method, as it takes the point cloud sampled from the 3D Bézier curve representation and encodes it into latent space, which is then used for optimization via Score Distillation Sampling (SDS). While other frameworks, such as Hunyuan3D[2], are potential candidates, many have only released the decoder code. Therefore, we selected TripoSG as it offers a publicly available encoder that fulfills this requirement and also demonstrates strong generation performance.
>
> [1] DiffSketcher: Text Guided Vector Sketch Synthesis through Latent Diffusion Models.
>
> [2] Hunyuan3D 2.0: Scaling Diffusion Models for High Resolution Textured 3D Assets Generation

---

> > ### Comment · Reviewer_X23k · 2025-08-05
> >
> > Thank you very much for the rebuttal. I think most of the concerns have been properly addressed. The responses to W1 to W3 and Q1 to Q3 are clear and convincing. However, I still have some further questions regarding W4.
> >
> > The authors mention that the salient edge points can be encoded and decoded, and reconstructed after adding and removing noise. I agree that TripoSG has strong encoding and decoding capabilities. But when it comes to generation, the key question is whether TripoSG reconstructs a 3D edge point cloud or a complete point cloud when given a sketch image as input.
> >
> > If it is the latter, it suggests that TripoSG tends to generate a complete point cloud from a sketch. In that case, SDS would be using a full point cloud rather than an edge-only point cloud to guide the 3D vector generation. Would this make TripoSG less suitable for guiding 3D vectors in this setting?
> >
> > It would be helpful if the authors could provide more details to clarify why the use of TripoSG here is appropriate.

---

> > > ### Author Response · Authors · 2025-08-06
> > >
> > > We thank the reviewer for the insightful comment and for giving us the opportunity to provide clarification on this matter. We would like to clarify the behavior of TripoSG under different input conditions. When given a sketch image, TripoSG outputs an extremely sparse mesh resembling a 3D wire sculpture, whereas a solid mesh is only generated when TripoSG is conditioned on an image containing object surfaces or solidity. This indicates that TripoSG essentially performs a mapping from sparse sketches to sparse edge point clouds, and from solid-appearance images to complete geometry.
> > >
> > > In our setting, we use images with object surfaces or solid appearance as input to TripoSG. Therefore, the latent representation encoded by TripoSG is optimized to reconstruct a complete point cloud or solid mesh, not just sparse edges. Although the Bézier curve-sampled point cloud differs in distribution from TripoSG’s training data, our empirical results indicate that the encoder remains robust—effectively encoding our inputs into meaningful latent codes. This supports the downstream SDS optimization and leads to improved final results.

---

> > > > ### Comment · Reviewer_X23k · 2025-08-06
> > > >
> > > > Thank you to the authors for the further clarifications. I believe all of my concerns have been clearly addressed. I have updated my score accordingly.

---

> ### Author Response · Authors · 2025-08-05
> **Welcome to discuss**
>
> Dear Reviewer X23k,
>
> Thank you again for your valuable feedback on our paper.
>
> We have submitted our rebuttal, which addresses the concerns you raised, including long strokes and the role of Primary Structure Fitting.
>
> As the discussion period concludes in two days, we would be grateful if you could take a moment to review our response and let us know if you have any further questions.

---

### Official Review · Reviewer_ztRZ · 2025-07-03

**Clarity:** 4
**Significance:** 3
**Originality:** 3
**Rating:** 4
**Confidence:** 4

**Summary:**

The paper presents a method to generate view-consistent 3D vector sketches based on an input image. The main idea of the method is to combine geometric fitting and 3D-prior guided refinement. The geometric fitting stage includes an image-to-3D model process, salient point cloud extraction, point cloud clustering, and Bézier curves fitting. This stage looks like an edge extraction process for a 3D model. In the 3D-prior guided refinement, a 3D SDS loss is used to guide the curve refinement process for regions overlooked in the first stage.

**Questions:**

1. Please clarify the technical novelty of the paper.
2. Please add comparisons with Dream3DVG [47].
3. Please discuss practical applications of the approach.

**Ethical Concerns:**

["NO or VERY MINOR ethics concerns only"]

**Final Justification:**

Dream3DVG is indeed a concurrent work, and it is not necessarily essential to this one; the practical applications of the approach are still relatively narrow, but as a research topic, it is acceptable. Therefore, I agree to accept this paper.

**Limitations:**

The practical applications of 3D sketches and the generation of them should be added and discussed.

**Quality:**

3

**Strengths And Weaknesses:**

# Strengths:
1.  The idea of combining traditional geometric fitting and 3D-prior guided refinement is interesting, and I think the former is key to maintaining 3D structure and view consistency.
2.  The writing is good. The paper is easy to read.
3.  The experiments are sufficient, including comparisons with existing approaches and ablation studies.

# Weaknesses:
1.  Although the idea of the paper is very interesting, the technical side looks straightforward and incremental by combining existing techniques.
2.  The comparisons with Dream3DVG [47] are lacking, as it also works on consistent views.
3.  I am curious about the practical applications of the proposed approach and the output results. The authors mentioned virtual reality creation in the introduction. I would like to see such applications and examples in the paper.

---

> ### Author Rebuttal · Authors · 2025-07-31
>
> ### **W1 & Q1: Novelty.**
>
> Our core contribution is the introduction of a **novel framework that directly addresses the fundamental issue of view-inconsistency** prevalent in existing 3D graphics generation methods. Previous works, such as 3Doodle[1], Diff3DS[2], and Dream3DVG[3], rely heavily on **2D constraints**: they project 3D gaphics into 2D views and utilize differentiable rendering to enforce alignment between the projections and image ground-truth. However, this reliance on 2D priors inherently limits their ability to enforce true 3D coherence across multiple views.
>
> In contrast, **our method introduces a paradigm shift by incorporating 3D priors directly into the optimization process for 3D vector graphics.** Specifically, we extract salient edges from the mesh and leverage a pretrained 3D generative model to define a 3D SDS loss in latent space, thereby completely removing the dependency on 2D projections. This enables optimization to be guided by a **holistic understanding of 3D shape geometry**, rather than by enforcing consistency across 2D views.
>
> To the best of our knowledge, **ViewCraft3D is the first framework to apply 3D SDS loss for optimizing structured 3D vector graphics**, which introduces both conceptual and technical novelty beyond a simple combination of existing components.
>
> ### **W2 & Q2: Comparison with Dream3DVG.**
>
> We should clarify that Dream3DVG is **concurrent work**, since its preprint was released (May 27) after the NeurIPS submission deadline. Therefore, the absence of this comparison in our initial submission is not a weakness. Nevertheless, we have conducted a quantitative comparison. Specifically, given that Dream3DVG accepts text prompts as input, we manually convert the input images from our quantitative experiments into text prompts and use them as input for Dream3DVG. As shown in the following table, our method achieves a higher aesthetic score than Dream3DVG.
>
> |                  | VC3D (Ours) | Dream3DVG |
> |------------------|-------------|-----------|
> | Aesthetic Score↑ | 4.352       | 4.1499    |
>
>
> In additional to the quantitative analysis, we also conducted a qualitative comparison between our method and Dream3DVG on two representative categories of objects. **Given the format restrictions that no addtional images can be added during the rebuttal phase, we describe the results below**:
>
> (1) For **man-made objects** with strong symmetry and straight-line dominance, such as cars, pistols, and houses, Dream3DVG often produces results resembling a 3D version of CLIPasso. **It tends to omit or oversimplify parallel structural lines representing object thickness.** For instance, vertical lines indicating the walls of a house are frequently missing. We hypothesize this is due to misleading optimization signals: parallel structural lines that should be generated simultaneously may appear overlapping or closely spaced in certain viewpoints, causing the optimization process to favor only one of them and ignore the rest.
>
> (2) For **natural objects** with asymmetric and richly curved structures, such as birds, butterflies, and flowers, **Dream3DVG suffers from more pronounced view inconsistency issues**. Taking the bird example (our result shown in Figure 1 of the main paper), Dream3DVG generates duplicated features from different viewpoints. For example, the beak and tail feathers appear twice, and an extra eye is hallucinated on the back of the head. These results demonstrate Dream3DVG's challenge in maintaining coherent and unified vector abstraction across diverse views of natural objects.
>
> These comparisons suggest that while Dream3DVG performs reasonably well on simple and rigid structures, it struggles with both the **geometric completeness** required for structured man-made objects and the **multi-view consistency** critical for deformable, natural categories. We will include the above results and discussion in the revised version.
>
> ### **W3 & Q3: Applications of Our Method and Resultant 3D SVGs.**
>
> The inherent editability of 3D vector graphics offers significant advantages for iterative modeling workflows, a potential explored in several academic works. (1) 3D vector curves can serve as **real-time guidance for surface creation and editing**. For instance, FiberMesh[4] enables the design of freeform surfaces by allowing users to directly draw and manipulate a network of 3D curves, which function as a structural vector scaffold for the surface. (2) Some works focus on reconstructing 3D surfaces from a network of 3D vector curves, treating the curve network as an **editable and interpretable representation that facilitates intuitive shape refinement**. Notably, True2Form[5] infers a complete 3D curve network from ambiguous 2D sketches, bridging the gap between sparse 2D input and structured 3D geometry.
>
> 3D vector graphics also support **animation production**. For example, Blender provides an official SVG plugin and keyframe interpolation tools. Users can project 3D vector models onto different viewpoints, and then generate animations using the plugin. This enables the creation of 2D animations from multiple angles while maintaining strong frame-to-frame consistency.
>
> 3D vector graphics are integral to VR/AR applications, and they have already been applied in some well-known software, such as Tilt Brush, Gravity Sketch, and Mental Canvas. With the rapid development of VR/AR devices and applications, 3D vector graphics will become more valuable. Our proposed approach can be used to generate 3D vector graphics assets.
>
>
> [1] 3Doodle: Compact Abstraction of Objects with 3D Strokes
>
> [2] Diff3DS: Generating View-Consistent 3D Sketch via Differentiable Curve Rendering
>
> [3] Empowering Vector Graphics with Consistently Arbitrary Viewing and View-dependent Visibility
>
> [4] FiberMesh: Designing Freeform Surfaces with 3D Curves
>
> [5] True2Form: 3D curve networks from 2D sketches via selective regularization

---

> ### Author Response · Authors · 2025-08-05
> **Welcome to discuss**
>
> We appreciate the reviewer's thoughtful engagement with our work. We believe the additional experiments and clarifications we have provided should address the concerns raised. We welcome any further questions or comments during the remaining discussion period and are committed to providing comprehensive responses.

---

### Note · Authors · 2025-08-14

We thank all reviewers for their thoughtful reviews, constructive questions, and active engagement. Your suggestions, proposed experiments, and requests for clarification have significantly improved our paper. We are pleased that nearly all reviewers indicated their concerns have been addressed.

The proposed ViewCraft3D directly addresses the challenge of view inconsistency in 3D graphics generation by introducing a novel paradigm: **integrating 3D priors directly into the optimization process**.

**Core Innovation:**
We present the first framework for generating structured 3D vector graphics that incorporates 3D priors at two levels: (1) **explicit geometry level**—extracting salient edges from the mesh, and (2) **implicit latent level**—defining a 3D Score Distillation Sampling (SDS) loss in latent space to leverage priors from a pretrained 3D generative model. This approach demonstrates clear superiority over both prior and concurrent methods. In detailed comparisons with the very latest work Dream3DVG (CVPR'25), ViewCraft3D not only achieves higher aesthetic scores but also qualitatively preserves geometric completeness in man-made objects and mitigates multi-view inconsistency in natural objects.

**Principled Framework:**
Our method’s robustness stems from a carefully designed two-stage optimization pipeline, rather than a simple combination of pretrained components. Stage I performs primary structure fitting, extracting and initializing salient geometric curves, which is indispensable for subsequent refinement. Ablation studies confirm that removing Stage I severely degrades quality. In response to Reviewer X23k's inquiries, we have supplemented ablations and clarified all technical details regarding clustering and point cloud sampling strategies.

**Broad Applicability:**
By focusing on the core challenge of view-consistent geometry, our framework produces lightweight, editable 3D vector graphics that can serve a wide range of downstream tasks. These include AR-based industrial assistance, CAD modeling, industrial design, and other applications where structured, editable 3D assets are valuable.

We believe ViewCraft3D delivers a clear, novel, and broadly relevant contribution to the NeurIPS community.

---

### Decision · Program_Chairs · 2025-09-17

**Decision:**

Accept (poster)

**Comment:**

This paper proposes a method to synthesize 3D Vector Graphics from a single input image. The key contribution of this paper lies in the proposed techniques for generating 3D parametric curves for a 3D mesh obtained via a pretrained image-to-3D generator. After the rebuttal, all four reviewers gave the same positive score (4: Borderline accept). Moreover, all reviewers mentioned clearly in the Final Justification that they intended to accept this paper. The AC has carefully checked the authors’ rebuttal, the reviewers’ comments, and the author-reviewer discussions. The AC agrees with the reviewers and suggests the acceptance of this paper. Please follow the suggestions of all reviewers to revise the manuscript.